# Advances and Highlights of miRNAs in Asthma: Biomarkers for Diagnosis and Treatment

**DOI:** 10.3390/ijms24021628

**Published:** 2023-01-13

**Authors:** Marta Gil-Martínez, Clara Lorente-Sorolla, Sara Naharro, José M. Rodrigo-Muñoz, Victoria del Pozo

**Affiliations:** 1Immunoallergy Laboratory, Immunology Department, Instituto de Investigación Sanitaria Fundación Jiménez Díaz (IIS-FJD), 28040 Madrid, Spain; 2CIBER de Enfermedades Respiratorias (CIBERES), Instituto de Salud Carlos III (ISCIII), 28029 Madrid, Spain; 3Department of Medicine, Faculty of Medicine, Universidad Autónoma de Madrid, 28029 Madrid, Spain

**Keywords:** asthma, miRNAs, epigenetics, biomarkers, phenotyping, endotyping, treatment

## Abstract

Asthma is a heterogeneous inflammatory disease of the airways that causes breathing difficulties, episodes of cough and wheezing, and in more severe cases can greatly diminish quality of life. Epigenetic regulation, including post-transcriptional mediation of microRNAs (miRNAs), is one of the mechanisms behind the development of the range of asthma phenotypes and endotypes. As in every other immune-mediated disease, miRNAs regulate the behavior of cells that shape the airway structure as well as those in charge of the defense mechanisms in the bronchi and lungs, controlling cell survival, growth, proliferation, and the ability of cells to synthesize and secrete chemokines and immune mediators. More importantly, miRNAs are molecules with chemical and biological properties that make them appropriate biomarkers for disease, enabling stratification of patients for optimal drug selection and thereby simplifying clinical management and reducing both the economic burden and need for critical care associated with the disease. In this review, we summarize the roles of miRNAs in asthma and describe how they regulate the mechanisms of the disease. We further describe the current state of miRNAs as biomarkers for asthma phenotyping, endotyping, and treatment selection.

## 1. Introduction

Asthma is a chronic respiratory disorder characterized by pathologic inflammation and tissue remodeling, causing reversible and variable airflow limitation and bronchial hyperresponsiveness. The main clinical symptoms of asthma are dyspnea, wheezing, cough, and chest tightness. This very common disease affects more than 300 million people worldwide and is associated with high morbidity and a great economic burden for health systems [1].

Asthma results from a complex interplay between genetic and environmental factors. Consequently, this very heterogeneous syndrome can be classified into multiple phenotypes and endotypes based on inflammatory cell profile, trigger type, or age of disease onset, among others [1,2]. 

One of the most common severe phenotypes is type 2 or eosinophilic asthma, which is characterized by the presence of eosinophils and the production of related type 2 cytokines. In allergic asthma, allergens are processed by antigen-presenting dendritic cells, which participate in the differentiation of naïve T cells into Th2 lymphocytes. Th2 cells produce several cytokines involved in asthma pathogenesis. Among these, interleukin-5 (IL-5) plays a crucial role in eosinophil differentiation, maturation, migration, and activation. IL-4 induces immunoglobulin E (IgE) class switching by B cells and is key to T cell differentiation into Th2 cells. IL-13, which shares a common receptor with IL-4 (IL-4Rα), also participates in isotype switching to IgE and induces goblet cell metaplasia and smooth muscle contractility. These cytokines act together to orchestrate type 2 inflammation. Most frequently, onset of allergic asthma occurs during childhood and manifests clinically as involvement of Th2 lymphocytes, presence of serum IgE, and/or a positive skin-prick test to given allergens. Specifically, IgE binds to the high-affinity IgE receptor (FcεRI) found on basophils and mast cells, leading to cell activation and the degranulation of inflammatory mediators (i.e., histamine, prostaglandins, and cysteinyl leukotrienes) [2,3,4]. 

In addition to the eosinophilic phenotype, type 2 asthma can be also non-allergic. Non-allergic asthma usually manifests later in life and shows no IgE reactivity to allergens or presence of Th2 lymphocytes. In this type, the triggers are microbes, contaminants, and glycolipids that produce epithelial damage, followed by alarmins such as thymic stromal lymphopoietin (TSLP), IL-25, and IL-33. These mediators stimulate type 2 innate lymphoid cells (ILC2s), which also produce large quantities of IL-5, IL-13, and IL-9, inducing type 2 inflammation [2,3].

Nowadays, it is very important to phenotype individuals with type 2 or eosinophilic asthma, as in recent years specific biological treatments have been developed for them. These biologics target eosinophils directly or indirectly by modulating the levels of different mediators implicated in this type of asthma [5].

Non-type-2 asthma is related to Th1 and Th17 immunity. This phenotype is also referred to as neutrophilic asthma because of the elevated presence of neutrophils in the blood and sputum of patients. In these individuals, there is an absence of type 2 cytokines. Treatment options for non-type-2 asthmatics are more limited, and these patients do not respond to corticosteroid drugs. In addition, the role of IL-17 and other related cytokines (i.e., IL-22) has not yet been fully elucidated in asthma, and more in-depth studies are needed to better characterize these patients. For example, a clinical trial using a novel anti-IL-17 receptor antibody (brodalumab; Amgen/AstraZeneca) showed minimal beneficial effects on asthma outcomes [6].

This distinction between T2 or eosinophilic asthma and Th17 or neutrophilic asthma is less clear-cut, and there is appreciable overlap in the types of cytokines found in both types of patients. Furthermore, due to this heterogeneity, patients respond differently to pharmacological therapies. In many patients, the disease can be controlled by inhaled corticosteroids and short- or long-acting β2-adrenergic agonists; however, in 5–10% of cases, the treatment is not enough, and these individuals progress to severe asthma. It is in these cases where biologic therapies are being applied. By way of example, omalizumab, an IgE-specific humanized monoclonal antibody, reduces serum IgE levels and asthma exacerbations, but has only mildly beneficial effects on forced expiratory volume in 1 s (FEV_1_). More recently, anti-IL-5 therapies have been developed. Firstly, anti-IL-5 humanized monoclonal antibodies, such as reslizumab or mepolizumab, decrease asthma exacerbations, and mepolizumab also reduces tissue and blood eosinophils in severe eosinophilic asthma. Secondly, a different mechanism has been found that causes benralizumab to deplete eosinophils through an anti-IL-5 receptor antibody with activation of ADCC (Antibody Dependent Cellular Citotoxicity) by NK cells, a treatment that controls asthma symptoms and improves lung function. Finally, dupilumab, which binds to IL-4Rα, blocks both the IL-4 and IL-13 pathways and improves asthma exacerbations and symptoms [4]. However, one of the side effects of this drug is that it can significantly increase the number of blood eosinophils, probably due to blockade of the IL-4/IL-13 pathway reducing eosinophil migration and causing blood eosinophils accumulation by inhibition of eotaxin-3, VCAM-1, and TARC without inhibiting eosinophilopoiesis in bone marrow [7,8,9,10]. New biologics are currently in the pipeline and will soon add new treatment possibilities, which will require more robust phenotyping/endotyping of patients.

To better classify asthma into different phenotypes and adapt treatments in these patients, epigenetic modifications have begun to be used in recent years, in particular micro-RNAs (miRNAs) employed as diagnostic and treatment biomarkers [11,12]. This review will offer an in-depth exploration of the role of miRNA regulation in asthma pathogenesis.

## 2. miRNA Regulation of Asthma Pathogenesis

As modulators of the immune response, microRNAs (miRNAs or miRs) regulate the messenger RNA (mRNA) of target genes and thus play a crucial role in the development and pathogenesis of asthma disease [13]. miRNAs are short, endogenous, single-stranded, non-coding RNAs typically consisting of 18–25 nucleotides; these evolutionarily conserved sequences regulate both expression of genes and gene networks and have been found to play a role as central regulators of post-transcriptional gene expression [14,15]. Direct binding of miRNAs to a target mRNA, which depends on the seven-nucleotide-long “seed sequence” of the miRNA in its 5′ region that base-pairs with complementary sequences within the 3′ untranslated regions (UTRs) of target mRNAs, may result in either mRNA degradation or translational repression of the target mRNA, leading to eventual gene silencing; it is estimated that more than 60% of human mRNAs have at least one conserved miRNA-binding site [16,17]. Further, it has been widely reported that a single miRNA can regulate many different mRNA targets by this mechanism of RNA interference (RNAi), and conversely, several miRNAs can cooperatively regulate a single mRNA target [15,18]. It should be noted that these small molecules can regulate the immune response through epigenetic mechanisms; although, like other RNAs, miRNAs have epitranscriptomic modifications that can alter their functionality, regulation, and biogenesis [19].

It has been speculated that miRNAs may be associated with the regulation of almost all aspects of cell physiology and are important to the survival and function of various types of immune cells. Additionally, miRNAs have been implicated as transcriptional regulators of a variety of biological processes [20,21]. This property makes miRNAs potential candidates for the management of immunity and disease control [20,22]. 

Since miRNAs are involved in post-transcriptional regulation of different mRNAs, their roles often depend on the interaction of miRNAs with their gene targets, the subcellular location of miRNA expression, the cell type, the abundance of miRNAs and target mRNAs, and their interaction affinity [23]. Interestingly, miRNAs can be tissue-specific and are able to contribute to the mRNA enrichment of the targeted tissue [24], while there are hundreds of miRNA pairs or combinations that achieve better repression of target mRNAs [25].

Although miRNAs are not new players in the study of disease development, their role as regulatory molecules in asthma is a relatively recent area of focus. Since miRNAs are extremely stable in a variety of body fluids, they are more amenable to non-invasive studies as molecules involved in the control of pathology mechanisms [26,27]. Several non-invasive techniques can be utilized, including next-generation small RNA sequencing (NGS) techniques, miRNA arrays, qPCRs, and digital-PCRs, with more in development and others attaining less popularity. The methodological variability and lack of inter-laboratory protocol homogenization and standardization may yield different results. Indeed, the fact that each approach has its limitations currently restricts progress in this field of study [28].

### 2.1. miRNA Control of Asthma Mechanisms

There is increasing evidence that miRNAs are involved in regulating disease mechanisms in asthma. Here, it is important to differentiate between two kinds of miRNAs: intracellular and extracellular (found inside extracellular vesicles) [29]. Vesicle-enclosed miRNAs are key paracrine messengers that mediate intercellular communication, and it has been described that cells are capable of performing selective loading of miRNAs into extracellular vesicles depending on the conditions or stimuli, for example, in hypoxic situations [30].

Studies in asthma have identified numerous miRNAs that may help to identify and better understand the disease, as compiled in other reviews of miRNAs in asthma and allergy [31]. MiR-221 blockade resulted in a reduction in the airway inflammation associated with asthma in an ovalbumin (OVA)-induced murine model of asthma [32]. It was suggested that miR-221 promotes the degranulation of mast cells and cytokine production [33]. Other works have reported that miR-21, produced in large amounts by macrophages and dendritic cells, is able to regulate T-helper (Th)1 and Th2 balance [34,35]. miR-21 also contributes to IL-4 expression, since in vitro re-stimulated miR-21-null mouse CD4+ T cells produced a higher level of IFN-γ and less IL-4 than wild-type cells [36]. Moreover, treatment with IL-13 up-regulates miR-21 in the epithelial cells of the human respiratory tract [34,37], and GM-CSF-treated eosinophils elevate expression of miR-21, which in turn promotes IL-13 action [38]. Previous investigations have shown that IL-13 is the direct target of miR-let-7, and it was also found that Th1 had significantly enhanced expression of miR-let-7a compared with Th2 cells [39,40]. 

In experimental asthma models with house dust mites, a greater level of miR-145 was observed in the airway wall [41]. Studies have documented that anti-miR-145 and anti-miR-126 significantly decrease allergic inflammation, eosinophil infiltration, mucus production, Th2 cytokine production (e.g., IL-5 and IL-13), and airway hyperreactivity [41,42,43]. miR-126 has been proposed as a component of innate immunity in response to an allergen and promotes Th2-mediated allergic inflammation [43]. Moreover, it was demonstrated that inhibition of miR-155 results in an increase in transcription factors implicated in the generation of a Th2 cell microenvironment [44,45]. In mouse lymphocyte studies, miR-146 was identified as being involved in determining Th0 differentiation into the Th1 or Th2 pathway [46]. In another article, expression of miR-146a and miR-146b was induced in respiratory tract smooth muscle cells from asthmatic patients after treatment with a cytokine cocktail, and these miRNAs negatively regulated COX-2 and IL-1β in these cells [47]. Furthermore, it has been postulated that down-regulation of miR-146a may partially mediate the severe asthma phenotype, as T cells deficient in miR-146a have been revealed to be over-activated in both acute and chronic inflammatory states [48]. 

In addition, miR-19a, another miRNA often altered in asthma, has multiple roles in the asthmatic airway, including modulation of remodeling through protein arginine methyltransferase 1 in smooth muscle cells [49]. Elevated miR-19a expression in airway T cells promoted type 2 cytokine production through direct targeting of PTEN and TNFAIP3, and the reduced miR-19a in airway smooth muscle cells led to increased airway remodeling [50]. Other research described a novel regulatory pathway involving miR-19a, which is critical to the severe asthma phenotype, in which down-regulation of miR-19a expression could be explored as a potential new therapy to modulate epithelial repair in asthma by targeting the *TGFβR2* gene [51]. We recently reported that miR-185-5p, a miRNA previously associated with asthma disease in serum, was able to control the synthesis of the matricellular protein periostin in airway structural cells, and could modulate the contraction of bronchial smooth muscle cells through the regulation of RhoA and CDC42, thereby making it a promising therapeutic target [52].

In another study, 4-fold lower expression of miR-20 was seen in alveolar macrophages from asthmatic mice compared to the control group [53]. In contrast, the level of VEGF expression in asthmatic mice was significantly elevated relative to control mice. Inhibition of miR-20b in alveolar macrophages was performed to test whether VEGF is down-regulated by this miRNA. The result was a gain in VEGF expression compared to control alveolar macrophages. However, after transfecting alveolar macrophages of asthmatic mice with miR-20b, VEGF expression was diminished. These data indicate that VEGF from alveolar macrophages is required for allergic airway inflammation in asthmatic mice and that miR-20b negatively regulates expression of VEGF [53]. Finally, an inverse correlation between lung function parameters and miR-16 in asthma has recently been reported. In silico analysis predicted that ADRB2, which is involved in bronchial smooth muscle contraction, was a target gene for miR-16, and this finding was later confirmed by luciferase assay [54]. 

Having demonstrated the alteration of a broad set of miRNAs in asthma disease, as exemplified above, and as depicted in Figure 1, further research is required to mechanistically define and fully comprehend their role(s) in human asthma pathogenesis.

### 2.2. miRNAs in Viral Exacerbations in Asthma

Asthma exacerbations are episodes in which there is worsening of asthma symptoms and lung function (Global Initiative for Asthma (GINA)) [55]. Many viral acute respiratory infections (ARIs) are thought to be the major triggers of exacerbation of symptoms in chronic respiratory diseases, especially in asthma; as a result, these ARIs are responsible for a substantial proportion of asthma-related morbidity and all asthma-related mortality [56]. It is well-documented in other reviews on the subject that miRNAs simultaneously carry out a regulatory function in controlling the immune response against respiratory viruses and those that cause viral ARIs, such as human rhinovirus (hRV), influenza virus (IV), human metapneumovirus (hMPV), human coronavirus (hCoV), and respiratory syncytial virus (RSV) and at the same time are modulated by these viral ARIs [57]. 

Consequently, alterations in the miRNA expression profile of cells associated with airway inflammation during viral infection suggest that miRNAs contribute to the pathogenesis of asthma disease exacerbations [58] (Figure 2). The expression of miRNAs during respiratory infections has attracted increasing attention in recent studies due to their potential as antiviral drug targets for ARI and as biomarkers for diagnosis and prognosis [59,60]. Bioinformatics tools have been very helpful in predicting in silico whether selected miRNAs target viral sequences, causing a favorable or opposing response to replication and, thus, survival of the virus [61]. 

When infecting human bronchial epithelial cells (HBECs), the respiratory viruses mentioned above activate the NF-κB and interferon signaling pathways to induce cell responses, limit viral replication, and thereby prevent tissue damage. It has been proposed that HBECs from asthmatic patients may have attenuated interferon responses, leading to increased viral shedding, greater activation of NF-κB signaling, and asthma exacerbations [62]. Thus, it can be expected that miRNAs targeted to the NF-κB pathway and that affect interferon signaling may have great capacity to modulate cellular responses to respiratory viruses and impact asthma exacerbations.

MiR-128 and miR-155 were identified as candidate regulators of the innate immune response in defense of hRV-1B, as they target hRV genetic material [62,63]. Gene silencing of these two miRNAs augmented hRV replication by up to 50% [62]. Another paper reported that miR-18a, miR-27a, miR-128, and miR-155 were down-regulated in asthmatic HBECs and that suppression of these four miRNAs simultaneously caused a significant rise in IL-8 and IL-6 expression [64]. Another miRNA involved in the immune response to hRV is miR-23b, which is implicated in down-regulating the expression of the transmembrane receptors LPR5 and VLDLR, which are crucial for endocytosis [65]. Finally, miR-122 is induced in epithelial cells under RV infection, its target being SOCS1, which is a suppressor of inflammation, thus promoting RV-induced lung disease in infants with severe bronchiolitis [66]. 

A current report evidenced up-regulation of miR-22 and down-regulation of its target genes HDAC4 and CD147 in response to IV A virus H1N1 in HBECs from healthy subjects. However, HBECs from asthmatic patients were unable to up-regulate miR-22 and displayed elevated and unaltered levels of HDAC4 and CD147, respectively [67]. A significant finding from other research is that down-regulation of miR-146a suppresses IV A replication by boosting the type I interferon response via its target gene TRAF6, both in vitro and in vivo, indicating that this miRNA could be a potential therapeutic target [68]. 

Moreover, miR-let-7f, an hMPV-induced miRNA in A549 cells, is functionally one of the most important miRNAs for viral replication control, presumably targeting hMPV RNA polymerase. The miR-let-7f inhibitor significantly enhances hMPV replication and progeny virus production, while miR-let-7f overexpression leads to an inverse effect [57,69]. Cells infected by hCoV switch on signaling cascades, which result in an elevation of NFKB1 and miR-9 expression. NFKB1 mRNA is targeted by miR-9, and this has been demonstrated to lead to the loss of NF-κB translation; nevertheless, this event is prevented by the OC43 hCoV nucleocapsid protein, which attaches to miR-9, permitting NF-κB translation [70].

Lastly, the three miRNAs miR-24, miR-124a, and miR-744, which belong to different families, intervene in the p38 MAPK pathway, exerting their effect on the kinases MK2 and Myc. These two kinases are key proviral factors, and their suppression by the three miRNAs mentioned above may provide antiviral protection against RSV [71].

Finally, application of novel methodologies that employ in vivo animal models, primary respiratory epithelial cell cultures, air–liquid interface cultures, and clinical samples will make it possible to further elucidate the functions of miRNAs during ARIs and during virally induced asthma exacerbations, opening the field of miRNA research in unexpected and promising ways [72].

## 3. miRNAs Are Promising Disease Biomarkers of Asthma and Its Phenotypes/Endotypes

As mentioned previously, miRNAs have been widely described as disease biomarkers for numerous diseases and conditions, owing in part to their biochemical and associative properties. The core features of disease biomarkers are differential expression in particular disease conditions and the ability to be detected easily and resist degradation. Indeed, miRNAs fulfill all these criteria due to their stability and disease-related expression in biofluids, as they travel inside vesicles and are bound to proteins in tissues, performing roles in post-transcriptional regulation [73]. 

Studies in the field of asthma biomarkers have searched for molecules or profiles that could be used to diagnose asthma in easily obtainable biofluids such as blood (serum and plasma), or from non-invasive samples that have a direct relationship with the airways such as sputum, bronchoalveolar lavage fluid (BAL), or exhaled breath condensate (EBC). Regarding the source, blood is easily and more routinely obtained, but airway samples can be used to relate miRNA expression to the pathophysiology of the disease.

Starting with the airways, Gomez et al. applied weighted gene correlation network analysis and were able to associate modules of miRNAs in sputum, including miR-223-3p, which were associated to hospitalizations, impairment of lung function, neutrophil frequency in sputum, and TLR/Th17 signaling [74]. Interestingly, the relation between miR-223-3p and sputum neutrophil count was described in another asthmatic cohort, which showed that miR-629-3p (derived from epithelial cells) plus miR-223-3p and miR-142-3p (from neutrophils) were increased in sputum from severe asthmatics, while also demonstrating in vitro that miR-629-3p regulates IL-8 and IL-1β in bronchial epithelial cells [75]. Moreover, increased expression of miR-145 was found in the sputum of asthmatics and in subjects with chronic obstructive pulmonary disease (COPD) compared with healthy individuals, while miR-338 expression was higher in asthmatic sputum compared with patients with COPD [76]. Finally, expression of miR-221-3p was decreased in the sputum, serum, and airway epithelium of asthmatic subjects, and the levels of this miRNA correlated with airway eosinophils. Artificial overexpression of miR-221-3p caused allergen-dependent eosinophil increase through CCL24, CCL26, and POSTN [77]. Regarding miRNAs in BAL, a profile of 16 exosome-derived miRNAs, including miRNAs belonging to the let-7 and miRNA-200 families, were differentially expressed in mild asthmatics, qualifying these as asthma biomarkers with a predictive power of 72% [78].

EBC is also a source of miRNAs for use as asthma biomarkers, as evidenced in a study in which six miRNAs including miR-1248, miR-21, and Let7a were found to be differentially expressed between healthy individuals, asthmatics, and patients with COPD [79]. Similarly, though in exosomes from EBC, a profile of 11 miRNAs, including miR-1246, miR-421, miR-595, and miR-624 exhibited differential expression in asthmatics compared with healthy subjects [80]. Regarding childhood asthma, miR-126-3p, miR-133a-3p, and miR-145-5p were positively associated with asthma; in contrast, miR-21 and miR-155-5p correlated negatively with symptomatic asthma and bronchodilation, while miR-146-5p was related to higher bronchopulmonary reversibility [81]. In a study from the same Portuguese group, the authors found that EBC miRNAs previously related to asthma, such as miR-133a-3p, were modulated in children by a high dietary acid load, providing a possible biomarker linking asthma and this type of dietary imbalance [82].

As blood samples are commonplace in clinical practice, samples including plasma and serum are the most widely used for clinical diagnosis and, thus, have been targeted for asthma-related biomarker discovery. One example is miR-21, a widely studied miRNA reported to be elevated in the serum of children with asthma and eosinophilic esophagitis (EoE) [83]. Similarly, serum miR-21 and miR-155 were found to be higher in asthmatics, serving as biomarkers for this disease with a specificity and sensitivity of over 95% [84]. Further, in childhood asthma, serum profiling found that the combination of miR-106a-5p, miR-18a-5p, miR-144-3p, and miR-375 could be used to discriminate asthma patients from healthy subjects with an area under the curve (AUC) of 0.94 [85]. Although many studies have found differentially expressed miRNAs among the asthmatic population, others have observed that miRNAs in asthma are also age-dependent, with decreased miR-106a and miR-126a found in elderly asthmatics, and lower miR-146a, miR-126a, miR-106a, and miR-19b in elderly controls [86]. Interestingly, asthma-related miRNAs such as miR-1165-3p are also characteristic of other related diseases such as allergic rhinitis (AR) or allergic bronchopulmonary aspergillosis (ABPA), thereby suggesting that they are versatile biomarkers [87].

Although miRNAs from individual blood samples could serve as biomarkers, their diagnostic power increases when they are combined into profiles. Panganiban et al. found that miR-1248, miR-26a, Let-7a, and Let-7d were differentially expressed in asthmatics compared to controls, and as such are specific for Th2 regulation [88]. Using plasma samples, the same group published another study in which random forest classification was used to build a prediction model, which led them to discover that miR-125b, miR-16, miR-299-5p, miR-126, miR-206, and miR-133b levels could be used to classify subjects into healthy individuals, asthmatics, and subjects with AR. Given the AUC value of 0.97 found by the authors, this profile is highly promising because of its diagnostic value [89]. Another study in plasma showed that miRNA ratios from human and murine samples could also be applied as biomarkers, depicting a profile of seven miRNA ratios that could differentiate asthmatics from healthy subjects with an AUC of 0.92 [90].

With this in mind, our research group also applied both logistic regression and the random forest bioinformatic method with distinctive eosinophil miRNAs evaluated in serum, and we observed that the combined expression of miR-320a/b, miR-185-5p, and miR-144-5p was able to classify subjects into asthmatics, with a further ability to sub-phenotype subjects by disease severity with high specificity (0.87) [91]. Interestingly, we also observed that some of these miRNAs, such as miR-1246 and miR-144-5p, are also increased in other respiratory diseases such as COPD and asthma–COPD overlap (ACO), thus indicating that they may be used as inflammatory respiratory disease biomarkers; others, such as miR-320a, are asthma-specific, and could therefore be used to differentiate asthmatics from other similar respiratory diseases—doing so with a high AUC value (over 0.84) [92]. 

A similar approach also recently found that the combined expression of miR-21-5p, miR-126-3p, miR-146a-5p, and miR-215-5p derived from serum exosomes was able to differentiate asthma disease severity using a logistic regression model; remarkably, some of these miRNAs were associated with type 2 high atopic asthma (miR-21-5p and miR-126-3p), while others were related to the IL-6^high^ endotype, or associated obesity, or neutrophilic asthma [93]. Our research group also tested miRNAs as biomarkers for differentiation of immunological phenotypes of asthma, finding that serum miR-26a-1-3p and miR-376a-3p were differentially expressed in eosinophilic asthmatics, as miR-26a-1-3p inversely correlated with blood eosinophil counts, and miR-376a-3p with FeNO and number of exacerbations [94].

Together, these results highlight the potential of miRNAs for use as diagnostic biomarkers for differentiating asthma diagnosis from healthy patients, identifying asthma versus other similar respiratory diseases, and for endotyping and phenotyping asthmatics in order to better understand the molecular pathways for optimal targeted therapy selection, as summarized in Table 1.

## 4. miRNAs and Their Relation to Asthma Treatments and Optimal Response

Due to the inherent heterogeneous characteristics of asthma, chronic treatment is generally required to reduce the symptom burden and minimize exacerbation. As such, asthma treatment is based on a stepwise strategy consisting of cycles of assessment, adjustment, and evaluation of the response to the selected treatment [95]. In most cases, mild to moderate asthmatics are controlled by the use of inhaled corticosteroids (ICS), which reduce airway inflammation and ameliorate asthma symptoms. Sometimes, however, a rescue dose of oral corticosteroids is required to treat exacerbations of the disease, as daily oral corticosteroids are only recommended for specific cases of severe asthma. Other therapies aimed at controlling the disease include the use of leukotriene receptor antagonists (LTRAs), long-acting β2-agonists (LABAs), or these in combination with ICS/formoterol (formoterol being the preferred reliever agent for GINA) [55]. Relief medication for isolated moments during asthma bronchoconstriction treatments includes short-acting β2-agonists (SABAs) and short-acting muscarinic antagonists (SAMAs) [96]. Finally, in severe asthma cases, where the usual treatments are insufficient, biological drugs, most of them monoclonal antibodies directed against immunological-disease drivers, can be used to reduce both disease burden and the doses of corticosteroids needed to reduce adverse effects. As mentioned previously, these include anti-IgE (omalizumab) for allergic asthma; anti-IL4/13Rα (dupilumab) for T2 asthma; anti-TSLP (tezepelumab) directed to the epithelial derived cytokine for broader asthma patients; and drugs directed at reducing eosinophilic inflammation such as anti-IL5 (mepolizumab and reslizumab) or anti-IL-5Rα (benralizumab), which completely deplete eosinophils [7,97,98,99,100,101]. As observed, there are many treatment options. Continued efforts should be made to discover novel biomarkers that, when combined with the empirical value of clinical assessment, will most likely facilitate treatment selection.

Indeed, some studies have reported in-depth explorations of miRNAs as biomarkers or detectors of the effects of asthma therapy and response to asthma treatments. As early as 2016, it was described that miR-21 levels in serum were able to differentiate between steroid-sensitive and -resistant asthmatic children; with an AUC of 0.99 and very high specificity and sensitivity, this is a promising biomarker for treatment effectiveness [102]. Moreover, in serum from asthmatic children, expression of miR-155-5p and miR-532-5p predicted the response to ICS with an AUC value of 0.86 due to the effect of those miRNAs on dexamethasone-induced transrepression of NF-κβ [103]. Other miRNAs including miR-146b, miR-206, and miR-720, together with clinical exacerbation scores, can help detect asthma exacerbation risk in children, with an AUC of 0.81, aiding in treatment evaluation and escalation [104]. Similarly, miR-16 in serum was described as a biomarker for response to salmeterol therapy, with an AUC value of 0.99, and it was shown that miR-16 was able to regulate adrenoreceptor β-2 (ADRB2) expression, modulating response to inhaled β-agonists [54].

Interestingly, miRNAs are modulated after treatment with asthma-related drugs, as shown in reports where miR-146a was observed to be increased in asthmatic plasma, and was associated both with higher blood eosinophil counts and a need for higher doses of corticosteroids, as this miRNA is capable of synergizing with the anti-inflammatory effect of corticosteroids in vitro [105]. Indeed, it would be very interesting to evaluate miRNA expression in a population of asthmatics before and after initiation of ICS treatment. Faiz et al. used airway biopsies from subjects with COPD and reported elevated miR-320d during ICS treatment compared to placebo, as this miRNA reduced cigarette smoke extract-induced inflammation by NF-κβ inhibition [106]. Regarding this matter, we found that expression of miR-144-3p was increased in the serum and lung tissue of asthmatics, and that this expression augmented with corticosteroids, hence making it a biomarker for severe asthma treated with higher doses of corticosteroids (AUC = 0.74) [107].

Although biological treatments for asthma have only been in clinical practice for a few years, their effectiveness and safety make them very promising compounds; therefore, their interaction with miRNAs and possible use as biomarkers for response to biological treatments is of considerable importance. Our research group was the first to conduct in-depth research on this topic, and we reported that serum miR-338-3p increased after treatment with reslizumab or mepolizumab in patients who responded well to treatment, indicating that miR-338-3p is a promising biomarker for reslizumab or mepolizumab response in severe eosinophilic asthmatics [108]. Moreover, we showed that serum levels of miR-1246, miR-5100, and miR-338-3p are altered after eight weeks of benralizumab administration, and that a correlation exists between miR-1246 expression and blood eosinophil counts, meaning that these miRNAs could be used as biomarkers of early response to benralizumab in asthmatic disease [109].

All of these reports demonstrate the importance of evaluating miRNAs in biological fluids of asthmatics before and after initiation or continuation of the treatments used to control this disease, as can be seen in Table 2. 

Research has proven that miRNAs are promising biomarkers and have the potential to be used to classify asthmatics into phenotypes and/or endotypes for optimal treatment selection, as well as to predict a favorable treatment response. Doing so would help raise us to the next level of personalized medicine and clinical care of asthmatic patients.

## Figures and Tables

**Figure 1 ijms-24-01628-f001:**
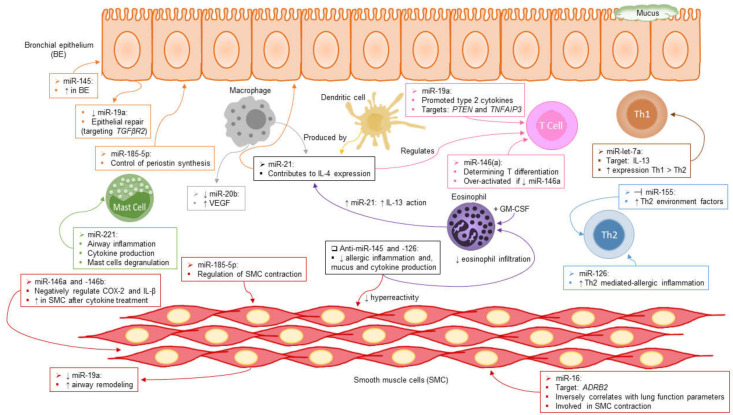
Role of miRNAs through the diverse pathophysiological mechanisms of asthma disease. miRNAs are able to regulate the functions of the airway structural cells, starting with bronchial epithelial cells, where miRNAs control remodeling, repair, and cytokine synthesis, and continuing with the smooth muscle compartment, as there are reports of miRNAs regulating contraction, hyperactivity, remodeling, and cytokine production. T cells (both Th1 and Th2) are also targets for miRNA-modulation in asthma, especially regarding polarization of immune responses toward the T2 axis. The behavior of eosinophils, the main effectors of severe asthma, is also regulated by miRNAs, altering their tissue infiltration capacity; meanwhile, other miRNAs control basophil degranulation and cytokine release. Finally, miRNAs are capable of tuning the actions of macrophages and dendritic cells, modulating their cytokine production above all.

**Figure 2 ijms-24-01628-f002:**
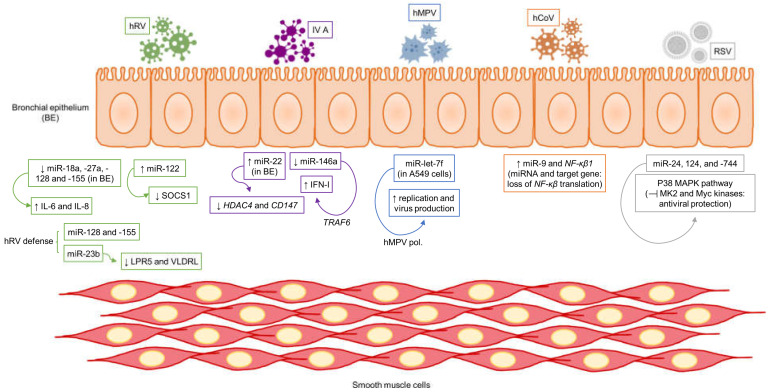
miRNAs are involved in virally induced asthma exacerbations. Several studies have shown the role of miRNAs in different mechanisms of viral exacerbations in asthma, most of which have sought to unravel how miRNAs control responses against the specific virus or how the virus interacts with host cells. Findings from this research reveal that miRNAs modulate antiviral immune responses, control virus replication, and determine how immune cytokine responses are orchestrated or regulated against respiratory viruses including human rhinovirus (hRV), influenza virus (IV), human metapneumovirus (hMPV), human coronavirus (hCoV), and respiratory syncytial virus (RSV). Sharp arrows indicate relationship or causation while blunt arrows signal inhibition.

**Table 1 ijms-24-01628-t001:** miRNA biomarkers of asthma disease and its phenotypes and/or endotypes.

miRNA Biomarker Profile	Clinical Value	Diagnostic Value	Biological Fluid	Reference
miRNA profile including miR-223-3p	Associated with sputum neutrophils, lung function impairment		Sputum	[74]
miR-629-3p, miR-223-3p, and miR-142-3p	Associated with sputum neutrophils		Sputum	[75]
miR-145 and miR-338	Differential expression in asthma or COPD		Sputum	[76]
miR-221-3p	Correlation with airway eosinophils		Sputum and serum	[77]
Sixteen exosome-derived miRNAs	Mild asthma biomarker	72% predictive power	BAL	[78]
Six miRNAs, including miR-1248, miR-21, and Let7a	Differentially expressed between healthy individuals, asthmatics, and COPD patients		EBC	[79]
Eleven miRNAs, including miR-1246, miR-595, and miR-624	Differential expression in asthma		EBC exosomes	[80]
Six miRNAs, including miR-146a-5p and miR-155-5p	Differential expression in childhood asthma and association with symptoms and bronchodilation		EBC	[81]
miR-133a-3p	Association between childhood asthma and high dietary acid load		EBC	[82]
miR-21	Differential expression in asthma and EoE		Serum	[83]
miR-21 and miR-155	Asthma	Over 95% sensitivity and specificity	Serum	[84]
miR-106a-5p, miR-18a-5p, miR-144-3p, and miR-375	Childhood asthma	AUC of 94%	Serum	[85]
miR-106a, miR-126a, and miR-19b	Asthma-related miRNAs are modulated by age		Serum	[86]
miR-1165-3p	Biomarker for allergic rhinitis and allergic bronchopulmonary aspergillosis (ABPA)	AUC of 70%	Serum	[87]
miR-1248, miR-26a, Let-7a, and Let-7d	Asthma related to Th2 immunity		Serum	[88]
Six miRNAs, including miR-16 and miR-126	Random Forest model for classifying patients as healthy, asthmatic, and allergic rhinitis	AUC of 97%	Plasma	[89]
Thirteen miRNA profiles	Asthma	AUC of 92%	Plasma	[90]
miR-320a/b, miR-185-5p, and miR-144-5p	Model for asthma disease and Random Forest severity classification	AUC over 75%	Serum	[91]
miR-1246, miR-320a, and miR-144-5p	Biomarker of asthma *versus* ACO and COPD	AUC of 84%	Serum	[92]
miR-21-5p, miR-126-3p, miR-146a-5p, and miR-215-5p	Asthma phenotyping and endotyping into T2 and non-T2 asthma	AUC of 89%	Serum exosomes	[93]
miR-26a-1-3p and miR-376a-3p	Association with blood eosinophilia and FeNO	AUC of 76%	Serum	[94]

AUC: area under the curve; ACO: asthma–COPD overlap; BAL: bronchoalveolar lavage; EBC: exhaled breath condensate; COPD: chronic obstructive pulmonary disease; EoE: eosinophilic esophagitis; FeNO: fractional exhaled nitric oxide.

**Table 2 ijms-24-01628-t002:** miRNAs and their relationships with asthma treatments.

miRNAs	Relationship with Asthma Treatment	Disease	AUC	Biofluid	Reference
miR-21	Corticosteroid sensitivity	Childhood asthma	99%	Serum	[102]
miR-155-5p and miR-532-5p	ICS responsiveness	Childhood asthma	86%	Serum	[103]
miR-146b, miR-206, and miR-720	Biomarker of exacerbation risk	Childhood asthma	81%	Serum	[104]
miR-16	Salmeterol therapy response and regulation of adrenoreceptor β-2 (ADRB2) expression	Asthma	99%	Serum	[54]
miR-146a	Corticosteroids and blood eosinophil counts	Asthma		Plasma	[105]
miR-144-3p	Corticosteroids treatment	Severe asthma	74%	Serum and lungs	[107]
miR-338-3p	Reslizumab or mepolizumab response	Severe asthma		Serum	[108]
miR-1246, miR-5100, and miR-338-3p	Early benralizumab response	Severe asthma		Serum	[109]

AUC: area under the curve; COPD: chronic obstructive pulmonary disease; ICS: inhaled corticosteroids.

## Data Availability

No new data were created or analyzed in this study. Data sharing is not applicable to this article.

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
