# Peer review of "Advances and Highlights of miRNAs in Asthma: Biomarkers for Diagnosis and Treatment"

_ijms, 2023, doi:10.3390/ijms24021628_

Round 1
Reviewer 1 Report
Line 401. ICS/Formoterol is the preferred reliever agent for GINA.
Line411. It was omitted tezepelumab which was approved by the FDA.
Author Response
REVIEWER 1
Comments and Suggestions for Authors
Line 401. ICS/Formoterol is the preferred reliever agent for GINA.
Answer: Dear reviewer, we have added formoterol as suggested, in line 411.
Line411. It was omitted tezepelumab which was approved by the FDA.
Answer: Thank you for the comment, as addressed, we have added tezepelumab and its use in lines 418-419.
Reviewer 2 Report
This is a very well written review of interest. My major concern however is the inadequate and incomplete referencing that must be corrected throughout the manuscript. Specific examples are given below but may not be fully comprehensive.
Comments:
1) line 91: remove "but does not....". One of the side effects of dupilumab is in fact an increase in blood eosinophils, probably related to the reduced expression of endothelial adhesion molecules under the regulation of IL4R alpha-signalling
2) The following references do not cite the original work but some sort of other article I believe that has eg reviewed and cited the original work. This must be corrected, eg: Cite 29, 30, 33, 37, 38, 40, 52, 60, 76 need to be replaced with the correct original work.
3) For the part on virus exacerbation, another interesting study should be mentioned and included into fig 2 that has experimental data on miR122 and data from infants with bronchiolitis ( Collison et al, JCI Insight 2021, doi: 10.1172/jci.insight.127933)
4) line 201, add the citation relevant here
Author Response
REVIEWER 2
Comments and Suggestions for Authors
This is a very well written review of interest. My major concern however is the inadequate and incomplete referencing that must be corrected throughout the manuscript. Specific examples are given below but may not be fully comprehensive.
Answer: We are grateful for the comments and for the corrections, indeed we have reviewed the complete manuscript in order to ensure that the references shown are linked to the original articles.
Comments:
1) line 91: remove "but does not....". One of the side effects of dupilumab is in fact an increase in blood eosinophils, probably related to the reduced expression of endothelial adhesion molecules under the regulation of IL4R alpha-signalling.
Answer: Dear reviewer, thank you for the comment, as suggested we have changed the sentences saying that “however, one of the side effects of this drug is that it can significantly increase the number of blood eosinophils, probably due to blockade of the IL-4/IL-13 pathway reducing eosinophil migration and causing blood eosinophils accumulation by inhibition of eotaxin-3, VCAM-1, and TARC without inhibiting eosinophilopoiesis in bone marrow” as seen in the manuscript lines 91-95.
2) The following references do not cite the original work but some sort of other article I believe that has eg reviewed and cited the original work. This must be corrected, eg: Cite 29, 30, 33, 37, 38, 40, 52, 60, 76 need to be replaced with the correct original work.
Answer: Thank you for the correction. We have revised the references that you mentioned and the rest of the revised text changing the citation to the original published paper.
3) For the part on virus exacerbation, another interesting study should be mentioned and included into fig 2 that has experimental data on miR122 and data from infants with bronchiolitis (Collison et al, JCI Insight 2021, doi: 10.1172/jci.insight.127933)
Answer: We are grateful for this information; we have added it to the text (lines 265-268) and figure 2 as suggested.
4) line 201, add the citation relevant here
Answer: Following your correction we have added the reference also in this line, in the revised text line 206.
Round 2
Reviewer 2 Report
The authors have made all requested changes and this interesting manuscript is now improved as proposed.